# Prevalence, etiology, and transmission of fibropapillomatosis in Olive Ridley turtles at a mass-nesting colony in the Mexican Pacific

Elizabeth Labastida-Estrada[1,2☉], Karina Marisol Lugo-Trejo[3☉], Valentina Islas-Villanueva[4], Francisco Benítez-Villalobos[5], Federico Alberto Abreu-Grobois[2], Alejandro Oceguera-Figueroa [ID][1]*

**1** Laboratorio de Helmintología, Departamento de Zoología, Instituto de Biología, Universidad Nacional Autónoma de México, Mexico City, Mexico, **2** Laboratorio de Genética, Instituto de Ciencias del Mar y Limnología. Universidad Nacional Autónoma de México, Mazatlán, Sinaloa, Mexico, **3** Programa de Maestría en Ciencias: Ecología Marina, Universidad del Mar, San Pedro Pochutla, Oaxaca, Mexico, **4** Secretaría de Ciencia, Humanidades, Tecnología e Innovación (SECIHTI), Instituto de Genética. Universidad del Mar, San Pedro Pochutla, Oaxaca, Mexico, **5** Instituto de Recursos. Universidad del Mar, San Pedro Pochutla, Oaxaca, Mexico

☉ These authors contributed equally to this work.
* aoceguera@ib.unam.mx

## Abstract

Fibropapillomatosis (FP) is a widespread disease in sea turtles characterized by the development of internal and external tumors that hinder physiological and general functions. Chelonid alphaherpesvirus 5 (ChAHV5) is recognized as the causative agent and has been studied and characterized worldwide. Here, we evaluated the prevalence and severity of FP in olive ridley females (*Lepidochelys olivacea*) during the mass-nesting season from July 2022 to January 2023 at Playa Escobilla, Oaxaca, on the Mexican Pacific coast. Using fragments of the capsid protein (UL18) and DNA polymerase (UL30) genes, we assessed the molecular detection of viral DNA in tumors and healthy tissues from FP-affected and apparently healthy females. To explore the potential role of the marine leech *Ozobranchus branchiatus* as a vector, we also screened leeches collected from both FP-affected and healthy turtles for viral DNA. Additionally, we identified ChAHV5 genetic variants from UL18 and UL30 sequences obtained from turtles and leeches. The estimated FP prevalence was 1.05%. Among the 62 FP-affected females, 82% were classified as mild, 13% as moderate, and 5% as severe. Molecular detection using the UL30 marker achieved a 100% detection rate in FP-affected turtles and 91% in apparently healthy individuals. In contrast, viral DNA was detected in only 6% of the analyzed leeches. Based on UL18, we identified a novel variant exclusive to olive ridley turtles (UL18_haplotype 3), while six UL30 variants were recorded, with UL30_Var01 being the most abundant and widely distributed in the Pacific. The low detection rate of ChAHV5 in leeches does not support their role as vectors. Our findings provide critical evidence of latent

**Data availability statement:** All genetic sequences generated in this study have been deposited in the DNA Data Bank of Japan (DDBJ) under accession numbers LC899718–LC899775 and LC899376–LC899383.

**Funding:** This project was funded by the Programa de Apoyo a Proyectos de Investigación e Innovación Tecnológica (PAPIIT IN215722 and IN226525) of the Universidad Nacional Autónoma de México (UNAM), which provided support to A.O-F., and by the Investigadoras e Investigadores por México program of the Consejo Nacional de Ciencia y Tecnología* (CONACYT), which provided financial support to V.I-V. (Project No. 538). The Dirección General de Asuntos del Personal Académico de la Universidad Nacional Autónoma de México (DGAPA-UNAM) granted a postdoctoral fellowship to E.L-E., and CONACYT also awarded a grant scholarship (No. 812903; CVU: 1165342) to K.M.L-T. There was no additional external funding received for this study. The funders had no role in study design, data collection and analysis, decision to publish, or preparation of the manuscript.

**Competing interests:** The authors have declared that no competing interests exist.

viral presence and highlight the need for integrating molecular surveillance in nesting sites to improve our understanding of FP dynamics and its potential impact on sea turtle conservation.

## Introduction

Fibropapillomatosis (FP) is a neoplastic disease of epizootic proportions affecting wild populations of endangered sea turtles worldwide [1]. It is a debilitating disease characterized by the development of single or multiple fibroepithelial lesions or tumors in internal and external organs and tissues, including the carapace and plastron [2]. Internal tumors are commonly found in the esophagus, lungs, heart, kidneys, intestine, spleen, liver, and muscles. Within the oral cavity, neoplastic formations can be found in the oral canthus, glottis, hard or soft palate, pharynx, and tongue [3]. External tumors typically occur in the eyes, neck, front flippers, axillary region, rear flippers, inguinal region, and tail [3]. FP lesions are proliferative masses that range from 0.1 to 30 cm in diameter [2]. Based on their external morphology, they can be smooth, verruciform, sessile, pedunculated, multi-lobulated, or cauliflower-like, and their coloration is defined by the pigmentation at the site of infection [2,4].

Turtles affected by FP typically develop masses on the body surface that can grow large enough hindering locomotion, feeding, growth, reproduction, and vision [5]. This condition leads to general debilitation and subsequently, animals die of starvation and dehydration or from secondary infections from ulcerated masses [5]. Internal tumors can disrupt normal organ function; for example, heart and kidney dysfunction, respiratory compromise, and gastrointestinal obstruction have been identified as causes of death [6].

The first reports of FP were described in 1936 in captive green turtles *Chelonia mydas* Linnaeus, 1758 at the New York Aquarium, New York, USA [7]. Later, in 1938, external tumors were observed in free-ranging green turtles in Key West, Florida, USA [7]. This disease was not recognized as a significant threat to sea turtles until the 1980s, when its geographic distribution and incidence increased dramatically [8]. For example, FP prevalence, estimated from physically examined stranded turtles, was as high as 92% in Kaneohe Bay, Hawaii [8]. Although FP was initially detected exclusively in green turtles, it has now been reported in the remaining six sea turtle species worldwide [2].

Based on experimental transmission trails in green turtles, Herbst et al. [9] identified a virus as the etiological agent of FP. The first study using molecular techniques in tumor samples of green, loggerhead (*Caretta caretta* Linnaeus, 1758) from Hawaii and Florida, and olive ridley turtles (*Lepidochelys olivacea* Eschscholtz, 1829) from Costa Rica, found three closely related herpesviruses: Florida green turtle herpesvirus (GTHV-Fl), Hawaiian green turtle herpesvirus (GTHV-Ha), and Olive ridley turtle herpesvirus (ORTHV) [10].

So far, eight herpesviruses have been identified in various freshwater and sea turtle species, altogether known as chelonid herpesviruses, five of them restricted to sea turtles [2,11]. The chelonid alphaherpesvirus 5 (*Scutavirus chelonidalpha* 5 or ChAHV5) has been the primary focus of research since it is considered the etiological

agent of FP [2,12]. Although its causative role has been widely accepted, Koch's criteria for establishing the microbiological etiology of infection and disease remain unfulfilled due to the inability to culture this virus in vitro [2,9]. This omission has been largely solved with molecular techniques demonstrating a correlation between the presence of FP and ChAHV5 [2].

ChAHV5 belongs to the subfamily Alphaherpesvirinae and has a double-stranded DNA genome of approximately 132 kb, divided into two regions: unique long sequence (UL) and unique short sequence (US) [12,13]. Within the UL region, target genes such as capsid protein (UL18), Glycoprotein H (UL22), Glycoprotein B (UL27), and DNA polymerase (UL30) have been widely used in PCR-based detection and characterization [10,14,15].

Herbst et al. [16] characterized five ChAHV5 variants, four of them (A – D) for loggerhead, Kemp's ridley (*Lepidochelys kempii* Garman 1880), and green turtles from Florida and North Carolina, U.S., and one variant from Hawaiian green turtles. In their phylogenetic study, the authors distinguished two lineages corresponding to Atlantic and Pacific samples [16]. Later, Greenblatt et al. [17] identified four ChAHV5 variants that corresponded to the Atlantic Ocean (Florida and Barbados), west Pacific (Australia), mid-Pacific (Hawaii), and east Pacific (Costa Rica and California). The global phylogeny of ChAHV5 conducted by Patrício et al. [18], based on UL18, UL27, UL30, and UL34 DNA sequences, recovered four geographical groups: Eastern Pacific (San Diego, Costa Rica, and Mexican Pacific), Western Atlantic/eastern Caribbean (Florida and Barbados), Atlantic (Gulf of Guinea and Puerto Rico), and Mid-West Pacific (Australia and Hawaii). These studies revealed that sympatric sea turtle species share the same viral variant and that genetic divergence between viral strains broadly correlates more strongly with geographic areas than with host species [16–18].

Previous studies pointed out that the genetic variants of ChAHV5 might be correlated with differences in disease dynamics and manifestations of the infection [1,19]. For example, these findings suggest that viral strains may be a potential factor influencing the manifestation, prevalence, or decline of FP among host species, populations, or individuals [3,19]. The geographical distribution of viral variants found in different sea turtle populations has been associated with host foraging grounds, suggesting horizontal transmission of ChAHV5 between different sea turtle species sharing habitats [2,20]. Direct transmission of the ChAHV5 through mating or aggressive interactions between individuals has been proposed [2], however, a role for mechanical vectors such as ectoparasitic leeches, bladder parasites, barnacles, and blood flukes has also been suggested [21,22].

Like most herpesviruses, ChAHV5 infection begins with inoculation followed by a systemic spread when the virus is present in the blood (viremia), indicative of viral replication [12,20]. Viral DNA can be detected in tumors, tissue biopsies, and blood samples of infected hosts [15,20]. Several studies have detected high levels of viral DNA in healthy turtles [15,20,23–25].

At present, FP is the most significant infectious disease affecting sea turtles and represents a global conservation challenge due to its widespread and rapid transmission [26]. The impact of FP on sea turtle populations remains unclear, as there are few systematic studies on the prevalence or severity of the disease [3,27–38]. Moreover, the factors that could increase the severity of FP remain unknown. Some studies have indicated that environmental factors, along with individual characteristics such as growth rate, sex, hematology, or parasite infections, could serve as potential triggers [3,29].

For the olive ridley turtle, no epidemiological data is currently available to assess the impact of FP on populations [26]. The first case of FP in olive ridley turtles was reported in 1982 at Ostional, Costa Rica [30]. Following this initial detection in nesting females, several studies have evaluated FP prevalence on nesting beaches, reporting values of 4–10% in 1998, 1.2% in 2005, and 14% between 2010 and 2013 at Ostional, Costa Rica; 1.4% in 1997 at Playa Escobilla, Mexico; 0.019% in 2022 in Baja California Sur, Mexico; and 78% between 2010 and 2013 in Nicaragua [4,31–34]. Additional reports have documented FP tumors in turtles captured in marine environments and in stranded individuals [35,36].

In the Mexican Pacific, documented reports on FP prevalence are scarce. In recent years, studies have focused on detecting the presence of the disease in foraging grounds along the northwestern Pacific and Baja California Peninsula coasts [37–39]. However, there are few studies on females affected by FP in nesting sites [32,34,40]. Such is the case for

Playa Escobilla in Oaxaca, where mass-nesting or 'arribadas' occur annually from June to February [41]. Playa Escobilla is the most important mass-nesting site for the olive ridley population in the eastern Pacific [41]. In this nesting site, the first records of females with tumors associated with FP were reported in 1980, but it was not until 1997 that a systematic study was conducted, estimating a prevalence of 1.4% in 9,201 nesting females examined [32]. Quackenbush et al. [42] analyzed herpesvirus DNA sequences from several localities, including an olive ridley female from Oaxaca, recognizing two variants, from Mexico and Costa Rica [42].

The objectives of this study were: (i) to estimate the prevalence and severity of FP in nesting olive ridley females in Playa Escobilla, Oaxaca, Mexico; (ii) to identify the genetic variants of ChAHV5 in FP-affected olive ridley females; (iii) to determine whether ChAHV5 can be detected in clinically healthy turtles using molecular techniques; and (iv) to assess the potential relationship between the leech parasitism and transmission of ChAHV5 in olive ridley females.

## Materials and methods

### Ethical statements

Capture, sampling, and monitoring of olive ridley sea turtles were conducted under permits SGPA/DGVS/04938/22, SPARN/DGVS/00945/22 and SPARN/DGVS/06005/23 issued by the Dirección General de Vida Silvestre (DGVS) and the Secretaría de Medio Ambiente y Recursos Naturales (SEMARNAT), Mexico. Additionally, the project was approved by the Ethics Committee for Research and Teaching of the Institute of Biology, UNAM (approval number: CoÉTICA-2025-005).

### Prevalence and severity of fibropapillomatosis in nesting olive ridley females

Sample collection and field data were conducted in Playa Escobilla, Oaxaca, Mexico, a Protected Natural Area designated as a 'sea turtle sanctuary' located in the southeastern Mexican Pacific. The nesting beach extends 15 km between Cozoalpetec River (15°43'37.47"N, 96°45'35.09"W) and Tonameca River (15°40'53.70"N, 96°37'06.71"W) in the municipality of Santa Maria Tonameca.

Sampling effort, defined as the number of females examined per night in each arribada throughout the entire nesting season, was determined based on the estimated size of the female nesting population of the previous three nesting seasons (S1 Table). During each arribada in the period July 2022 to January 2023, night surveys were conducted along the nesting beach to detect FP in nesting females. Surveys were carried out on foot along the nesting area during the five nights of each arribada, between 21:00 and 04:00 h, covering approximately 7 km of beach. After completing oviposition, 384 females were randomly selected each night for physical examination. Each female was examined to determine the presence of tumors attributable to FP. We examined the dorsal and ventral body surfaces, focusing on the eyes, neck, shoulders, forelimbs, hind limbs, cloaca, axillary, and inguinal regions. All examined females were tagged with non-toxic, water-based traffic paint (Ecolatex®) to avoid resampling. For each arribada, the number of FP-affected females and the total number of females examined were recorded. The prevalence of FP was estimated according to Pfeiffer [43] for each arribada and the entire nesting season.

For each female examined, the number and distribution of tumors, as well as its physical characteristics (color and morphology) were recorded. In addition, photographs of each tumor were taken at 30 cm, at a 90° angle, with a metric reference (flexible tape or caliper), to later estimate its diameter using ImageJ software v. 1.54a (National Institutes of Health, NIH). Tumors were classified into four categories, based on approximate size according to Work and Balazs [28]: Category A (<1 cm), B (1–4 cm), C (4–10 cm), and D (>10 cm). To determine the disease severity in each affected female, the Fibropapilloma Index (FPI) was estimated according to Rossi et al. [3]. Based on the FPI score, each affected turtle was classified as mild (FPI < 40), moderate (40 ≤ FPI < 120), or severe (FPI ≥ 120) [3].

### Viral DNA detection in nesting olive ridley females

During the arribadas from August and October 2022, three kinds of tissues were sampled: (A) tumors, (B) healthy tissues from FP-affected turtles, and (C) tissue or blood from clinically healthy turtles (individuals with no visible signs of

FP). Tumor and flipper biopsies were obtained using a sterile disposable biopsy punch (diameter 6 mm, ~140 mg) for each individual and maintained in 96% ethanol in sterile Eppendorf tubes. Blood samples were collected from the dorsal cervical sinus following the protocol described by Dutton [44]. A total of 0.5 mL of blood was obtained from each individual and preserved in 96% ethanol at 4°C in sterile Eppendorf tubes. Sampling was conducted after oviposition was finished to avoid interfering with the nesting activities and minimize discomfort and stress to the turtles. To reduce the risk of infection, a topical antiseptic (povidone–iodine) was applied to the cutting site both before and immediately after tissue obtention. Genomic DNA was extracted from approximately 50 mg of tissue (tumors and healthy flipper tissue), and for blood samples, we used approximately 20 μL of whole blood using the DNeasy Blood & Tissue kit (QIAGEN) according to the manufacturer's instructions. Detection of ChAHV5 gene was performed by polymerase chain reaction (PCR) targeting partially conserved regions for DNA polymerase [14] (UL30; fragment of approximately 445 bp) and capsid protein gene [15] (UL18; fragment of approximately 140 bp). Amplifications of ChAHV5 partial gene regions were carried out in a 10 μL reaction volume containing 2 μL of DNA template (~20 ng), 1X GoTaq Buffer (PROMEGA), 2 mM MgCl$_2$, 0.2 mM of dNTPs, 0.2 μM of forward and reverse primers for UL30 and UL18 (S2 Table), and 1U of GoTaq DNA polymerase (PROMEGA). Thermal cycling conditions for PCR were an initial denaturation at 94°C for 5 min, followed by 35 cycles of 94°C for 60 s, an annealing temperature for 60 s, extension at 72°C for 90 s, and a final elongation step at 71°C for 2 min. Amplicons were visualized on 1.5% agarose gels in 1X TAE buffer using GelRed (BIOTUM) stain and a 100 bp Plus DNA ladder (VIVANTIS) as reference. Purified PCR amplicons were sequenced using an ABI PRISM 3730 sequencer (Applied Biosystems, Carlsbad, CA) at the Laboratorio Nacional de la Biodiversidad, Instituto de Biología, Universidad Nacional Autónoma de México (LANABIO-IB, UNAM).

Negative controls were included in each PCR batch, and positive amplicons were confirmed by repeating PCR. Positive amplicons were sequenced in both directions using the same primers used for amplification. Only the samples that met these criteria were considered in the subsequent analyses. Forward and reverse sequences were analyzed and edited in GENEIOUS PRIME 2023.0.4 (BIOMATTERS Ltd., Auckland, New Zealand).

All DNA sequences obtained in this study were subjected to a BLASTn [45] search in the NCBI database to confirm the presence of viral DNA. Newly generated sequences had a percent identity > 99% with ChAHV5. The detection rate was estimated as the proportion of viral detection per PCR assay method, in total cases [15]. Detection rates were calculated across all sample groups collected.

## Detection of ChAHV5 in marine leeches

During the physical examination of olive ridley females, marine leeches were removed from the skin of turtles, both with FP tumors and healthy turtles. All specimens were preserved in 96% ethanol at 4°C for further molecular studies. Leeches were identified using taxonomic keys and specialized literature [46,47]. To confirm the taxonomic identity, we extracted Genomic DNA from each specimen using the Animal and Fungi DNA Preparation Kit (Jena Bioscience, GmbH, Germany) according to the manufacturer's instructions. Then, a fragment of 658 bp region of the mitochondrial COI DNA gene were sequenced using the primers LCO1490 and HCO2198 [48]. Amplifications of the COI gene fragment were carried in a 15 μL reaction volume containing 2 μL of DNA template (~20 ng), 0.2 μL of each primer, 1X of reaction Buffer, and 0.5 U My Taq DNA polymerase (Bioline, London UK). The thermal cycling conditions for PCR were an initial denaturation at 94°C for 5 min, followed by 30 cycles of 94°C for 45 s, an annealing temperature of 48°C for 45 s, extension at 72°C for 1 min, and a final elongation step at 72°C for 7 min. Amplicons were visualized on 1.5% agarose gels in 1X TAE buffer. PCR products were sequenced at LANABIO-IB, UNAM. The obtained sequences were analyzed as previously described and compared with the NCBI database using BLASTn [45] to confirm the taxonomic identity.

Molecular detection of ChAHV5 in marine leeches was performed by PCR assay using partially conserved regions for UL30 according to the protocol previously described for tumor tissue. Positive amplicons were sequenced at LANABIO-IB, UNAM, and sequences obtained were included in subsequent analyses.

### Identification of genetic variants of ChAHV5

Newly generated capsid protein UL18 (~118 bp) and DNA polymerase UL30 (~483 bp) sequences were aligned with sequences available in GenBank and Dryad repositories (S3 and S4 Tables). Additionally, we included partial regions of UL18 and UL30 genes from Testudinid alphaherpesvirus 3 (Te-HV3), isolated from *Testudo horsfieldii*, as an outgroup (GenBank accession number NC027916). Phylogenetic analyses for each locus were conducted using two optimality criteria: Parsimony and Maximum Likelihood (ML). Parsimony analyses were conducted in TNT v. 1.6 [49], using the New Technology search algorithms and branch support was assessed through bootstrap analysis with 10,000 replicates. ML analyses were performed in RaxML-HPC implemented in raxmlGUI 2.0 [50] using the following options: general time-reversible (GTR) model evolution, a gamma-distributed rate variation (G), and 1000 rapid bootstrap replicates.

To identify ChAHV5 variants obtained from olive ridley females, we used the SNP-sites tool [51] on alignments of the capsid protein (UL18) and DNA polymerase (UL30) sequences. Polymorphisms were extracted in Variant Call Format (VCF) and visualized using VCF-Explorer [52].

## Results

### Prevalence and severity of fibropapillomatosis in nesting olive ridley females

A total of 6,660 nesting females were examined during seven arribadas from July 2022 to January 2023. Seventy females (1.05%) presented tumors compatible with FP (Table 1). Of the females with evidence of FP, disease severity was evaluated in 62 cases (S5 Table). In total, 232 tumors were recorded, with the number of tumors per individual ranging from 1 to 30. The anatomical areas affected by tumors were the forelimbs (N = 88), shoulders (N = 80), and neck (N = 57). Most tumoral masses fit into categories B (>1–4 cm) and C (>4–10 cm). Nine tumors larger than 10 cm in diameter were recorded. Based on the FPI classification, 51 cases were mild (82%), eight cases were moderate (13%), and three cases severe (5%) (Fig 1, S5 Table).

### Viral DNA detection in nesting olive ridley females

A total of 42 olive ridley DNA samples were obtained including 18 tumor biopsies, two samples of healthy flipper tissue from sick females, and 21 samples from clinically healthy turtles, comprising nine blood samples and 12 flipper tissue samples. Additionally, a necropsy was performed on one female found dead on the beach with no external and internal tumors associated with FP, however a muscle tissue sample was collected for molecular detection of ChAHV5.

Screening for ChAHV5 using PCR targeting the capsid protein gene UL18, the detection rate in tumor biopsies was 12 out of 18 females (67%) and in samples from clinically healthy turtles, two of 22 samples were positive (9%). No viral

**Table 1. Prevalence of FP-affected olive ridley females during the 2022-2023 nesting season in Playa Escobilla, Oaxaca, Mexico.**

| Arribada month | Estimated arribada size* | Examined females | FP-affected females | Prevalence (%) |
|---|---|---|---|---|
| July | 25,191 | 318 | 2 | 0.63 |
| August | 276,507 | 1532 | 20 | 1.3 |
| September | 243,535 | 1224 | 9 | 0.7 |
| October | 533,805 | 1084 | 7 | 0.6 |
| November | 293,331 | 1372 | 20 | 1.4 |
| December | 51,138 | 274 | 4 | 1.46 |
| January | 36,384 | 856 | 8 | 0.93 |
| **Overall** | **1,423,507** | **6,660** | **70** | **1.05** |

\* Data provided by the Arribadas estimation system from the National Sea Turtles Program (National Commission for Protected Natural Areas, CONANP 2023).

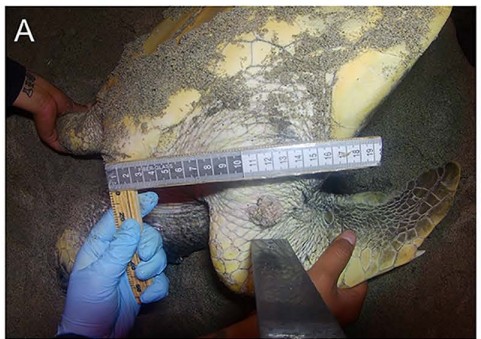
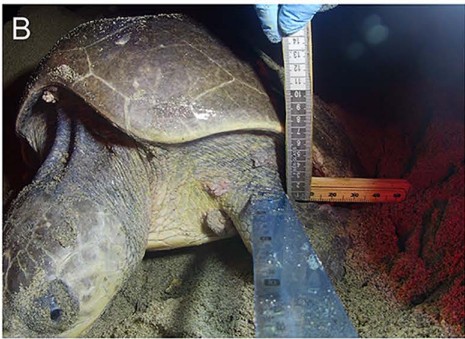
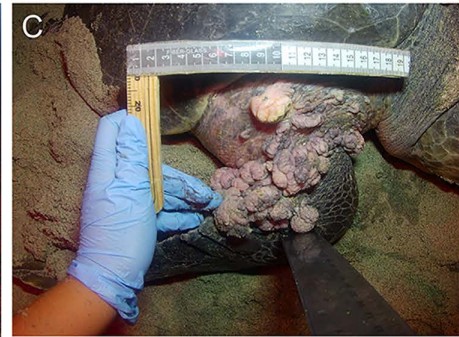

**Fig 1. Severity of FP cases in olive ridley females at Playa Escobilla, Oaxaca, Mexico, based on the Fibropapillomatosis Index (FPI).** (A) Mild, (B) Moderate, and (C) Severe.

DNA was detected in samples of healthy tissue from sick females using this locus. The overall detection rate for the UL18 locus was 33% (Table 2). Using the UL30 gene revealed that all 18 tumor biopsies were positive (100%). Two samples of healthy tissue from sick females were also positive (100%). Among clinically healthy turtles, ChAHV5 was detected in skin samples from 20 of 22 individuals (91%). When all samples were considered, the overall detection rate was 95% for UL30 locus.

### Marine leeches identification and detection of ChAHV5

All leech specimens presented seven pairs of lateral branchiae and a single pair of eyespots. Based on the information from literature, leech specimens were identified as *Ozobranchus branchiatus* (Menzies, 1791). Additionally, a DNA fragment of ~608 bp of COI of eight randomly selected specimens were generated. Identification using a BLASTn search confirmed all sequences as *O. branchiatus*.

Of the 33 leeches sampled from clinically healthy and the two from FP-affected turtles, only two specimens from clinically healthy turtles (6%) tested positive for the ChAHV5.

### Identification of genetic variants of ChAHV5

A total of 50 sequences of the UL18 gene representing various localities and hosts were obtained from GenBank and Dryad repository (S3 Table) and analyzed in combination with 14 sequences obtained in this study from olive ridley turtles. Parsimony analysis resulted in a single most parsimonious tree of 60 steps. ML analysis recovered a tree with a log-likelihood of −342.54. The parsimony tree (S1 Fig) grouped most sequences (13 of 14) of the UL18 gene obtained from olive ridley turtles at Playa Escobilla in a single clade (gray box). One sample obtained for this study was found to be nested within a different group (green box) that includes sequences obtained from hawksbill *Eretmochelys imbricata*

**Table 2. Percentage of detection of ChAHV5 in olive ridley females in Playa Escobilla, Oaxaca, Mexico using UL18 and UL30 genes.**

| Sample group | Size sample | Sample type | UL18 | | UL30 | |
|---|---|---|---|---|---|---|
| | | | N positive | % detection | N positive | % detection |
| FP-affected turtles | 18 | FP tumor tissue | 12 | 67 | 18 | 100 |
| Healthy tissue of FP turtles | 2 | Fin tissue | 0 | 0 | 2 | 100 |
| Clinically healthy turtles | 22 | Fin tissue, muscle or blood | 2 | 9 | 20 | 91 |
| **Overall** | **42** | **All sample types** | **14** | **33** | **40** | **95** |

Linnaeus, 1766, leatherback *Dermochelys coriacea* Vandelli,1761 and green turtles from various regions worldwide. Similarly, the ML phylogenetic tree (Fig 2) recovered most UL18 sequences obtained from olive ridley turtles at Playa Escobilla in a single group, within samples from the same host collected in the Caribbean, Portugal and Brazil. One sample from Escobilla appeared in a different part of the tree, with no relationships with the remaining samples.

A total of 97 UL30 sequences were obtained from GenBank, representing various localities and sea turtle species (S4 Table). These sequences were analyzed alongside 44 sequences obtained in this study: 42 from turtle tissue and two from *O. branchiatus* leeches. Parsimony analysis resulted in six equally parsimonious trees of 231 steps each. ML analyses recovered a tree with a log likelihood of −1551.12. The parsimony strict consensus tree (S2 Fig) grouped most sequences of the UL30 gene obtained from olive ridley turtles at Playa Escobilla and with sequences from the same host collected in Ecuador, Nicaragua, Costa Rica, Chile (gray box). Within the same clade, two viral sequences isolated from leeches –one from Escobilla, Oaxaca and one from Baja California– were also located. All these samples in this clade originated from the Pacific region. A second group (green box) included a variety of samples derived from green, loggerhead, leatherback and Kemp's ridley sea turtles *Lepidochelys kempii* Garman, 1880, primarily from the Atlantic Ocean, although samples from the Pacific (Costa Rica, Hawai, Australia and, surprisingly, from Escobilla, Oaxaca, Mexico) were also represented. In the ML tree (Fig 3), one sample derived from olive ridley turtle and one from a leech, both from Playa Escobilla, Oaxaca, were found nested within a large group of samples derived predominantly from green turtles from the Pacific. The remaining viral DNA sequences generated for this study did not exhibit any phylogenetic structure.

Analysis of the UL18 final matrix revealed nine variable sites, which defined five variants, two of these correspond to haplotypes 1 and 2 as defined by Alfaro-Núñez et al. [23] (Fig 4, S6 Table). In this study, we identified a novel variant exclusive to olive ridley turtles, designated as UL18_haplotype 3. This variant was the most frequent in samples from Playa Escobilla (13 of 14; 93%). The fourth variant, UL18_haplotype 4 (OQ189658), originally reported in green turtles from Mabul Island, Malaysia, was detected in one olive ridley individual from Playa Escobilla (1 of 14; 7%). Finally, UL18_haplotype 5 (GenBank accession number KY933583) was previously identified in green turtles from northern, central, and eastern Taiwan.

Exclusively for olive ridley turtles, we analyzed 56 UL30 gene sequences from samples collected in Mexico (Baja California, Sinaloa, and Oaxaca), Costa Rica, Nicaragua, and Chile. Additionally, three sequences obtained from *O. branchiatus* leeches were included: two from Playa Escobilla, Oaxaca (this study) and one from Baja California, Mexico. Analysis of the final matrix revealed 25 variable sites, defining six variants exclusive for olive ridley turtles (Fig 4, S7 Table). The most prevalent variant, UL30-Var01, originally characterized in Costa Rica (GenBank accession number AF049904), was the best represented, accounting for 49 of 56 samples (88%) and exhibiting a wide geographical distribution, from Baja California, Mexico to San Antonio, Chile, including samples from Nicaragua. This variant was dominant in olive ridley females from Playa Escobilla, being present in almost 98% of the samples (41/42) and detected in one leech collected from a clinically healthy turtle from Playa Escobilla. The second variant, UL30-Var02 (GenBank accession number AF209009), previously detected in a nesting female from Oaxaca, Mexico, was not detected in our study. The remaining four variants, UL30-Var03 (GenBank accession number MH450167; Sinaloa, Mexico), UL30-Var04 (GenBank accession number KP724838; Nicaragua), UL30-Var05 (KP724841; Nicaragua), and UL30-VarF06 (GeBank accession number KP724845; Nicaragua) were each represented by a single sample. Finally, the variant CmHA (GenBank accession number AY646893), originally reported in green turtles from Hawaii, was detected in this study in one olive ridley female (1 of 42, 2%) and one *O. branchiatus* leech from Playa Escobilla, Oaxaca.

## Discussion

### Prevalence and severity of fibropapillomatosis in nesting olive ridley females

Since the last prevalence of FP report for FP in female olive ridley turtles nesting at Playa Escobilla, published in 1998 [32], prevalence values have remained relatively constant. During this period, FP prevalence of 1.4% (9,201 turtles examined) in 1997 slightly decreased to 1.05% (6,660 turtles examined) in the present study. Notably, there are no

**Fig 2. Maximum likelihood phylogenetic hypothesis based on DNA sequences of the UL18 gene of ChAHV5.** Sequences obtained from olive ridley turtles at Playa Escobilla, Oaxaca, Mexico are indicated in green. Sample groups are coded as follows: A = FP-affected turtles, B = healthy tissue from FP-affected turtles, and C = clinically healthy turtles. Bootstrap values (≥50%) are indicated next to nodes. The tree is rooted with Testudine alpha-herpesvirus 3 (*Scutavirus testudinidalpha3*; GenBank accession number NC027916). For sequences retrieved from GenBank or the Dryad repository, the accession numbers or ID are provided in parentheses. Locality codes: are GG = Gulf of Guinea, BR = Brazil, CR = Costa Rica, DNK = Denmark, KWT = Kuwait, MY = Malaysia, MX = Mexico, TWN = Taiwan, U.S. = United States.

documented cases of FP in Playa Escobilla before the late 1980s [32], suggesting either that the disease was absent in the populations or that its prevalence was too low to be detected. Regardless of this, it is remarkable that over a period of approximately 25 years, the prevalence of FP has remained low and constant. The prevalence of FP in female olive ridley turtles from Playa Escobilla contrasts with that reported in a recent study conducted in 2022 on solitary nesting females

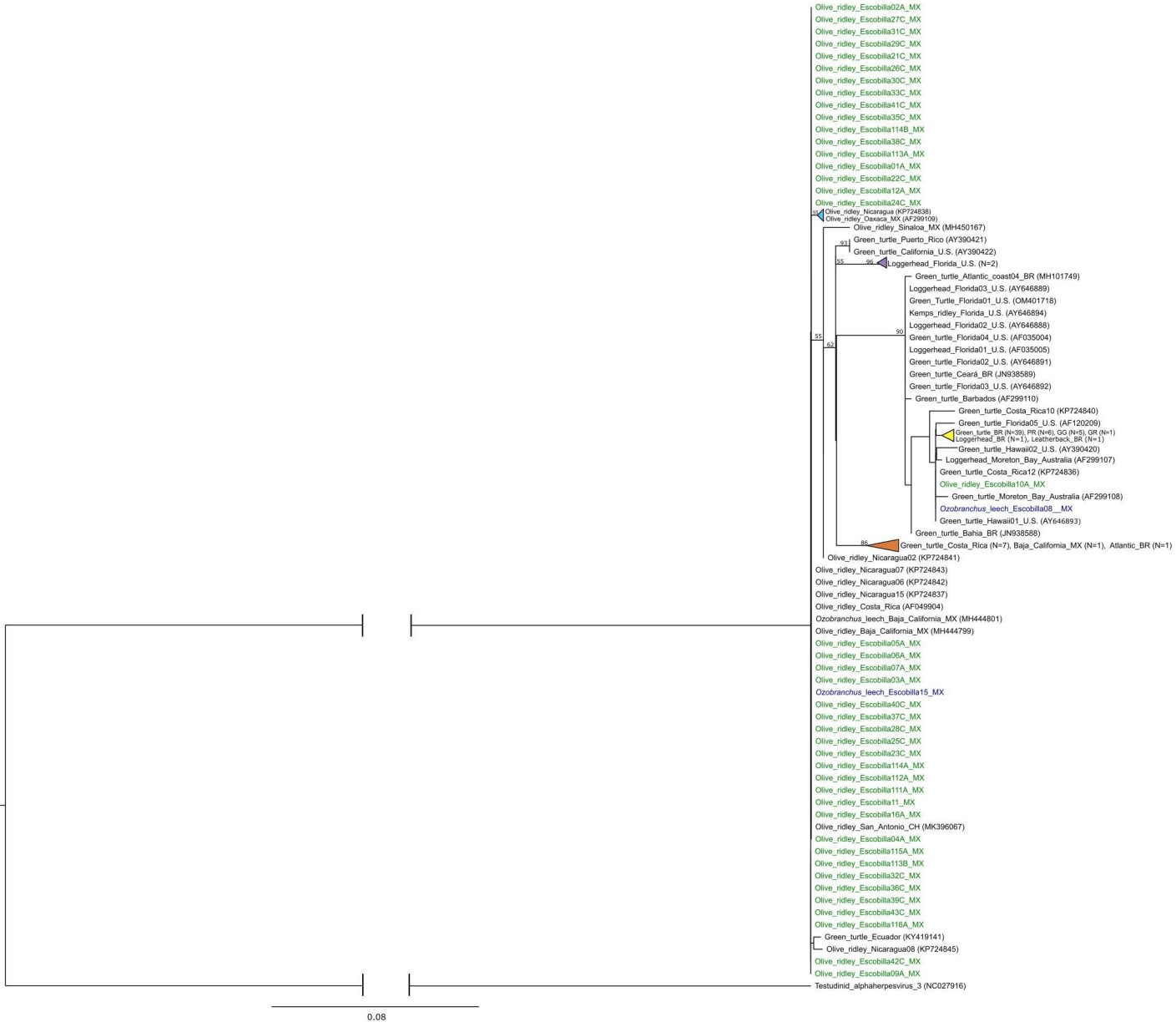

**Fig 3. Maximum likelihood phylogenetic hypothesis based on DNA sequences of the UL30 gene of ChAHV5.** Sequences isolated from olive ridley turtles at Playa Escobilla, Oaxaca, Mexico are indicated in green. Sample groups are coded as follows: A = FP-affected turtles, B = healthy tissue from FP-affected turtles, and C = clinically healthy turtles. Viral sequences obtained from the marine leech *Ozobranchus branchiatus* in blue. Bootstrap values ≥50% are shown next to nodes. The tree is rooted with Testudine alphaherpesvirus 3 (GenBank accession number NC027916). For sequences retrieved from GenBank, the accession numbers are provided in parentheses. Country codes: BR = Brazil, CH = Chile, GG = Gulf of Guinea, GR = Grenada, MX = Mexico, PR = Puerto Rico, and U.S = United States.

from Baja California Sur, Mexico, which report a significantly lower prevalence of 0.019% [34]. These results suggest that mass-nesting, a notable characteristic of this population, may facilitate the close contact among infected and healthy females for a short period of time, thereby promoting the virus transmission.

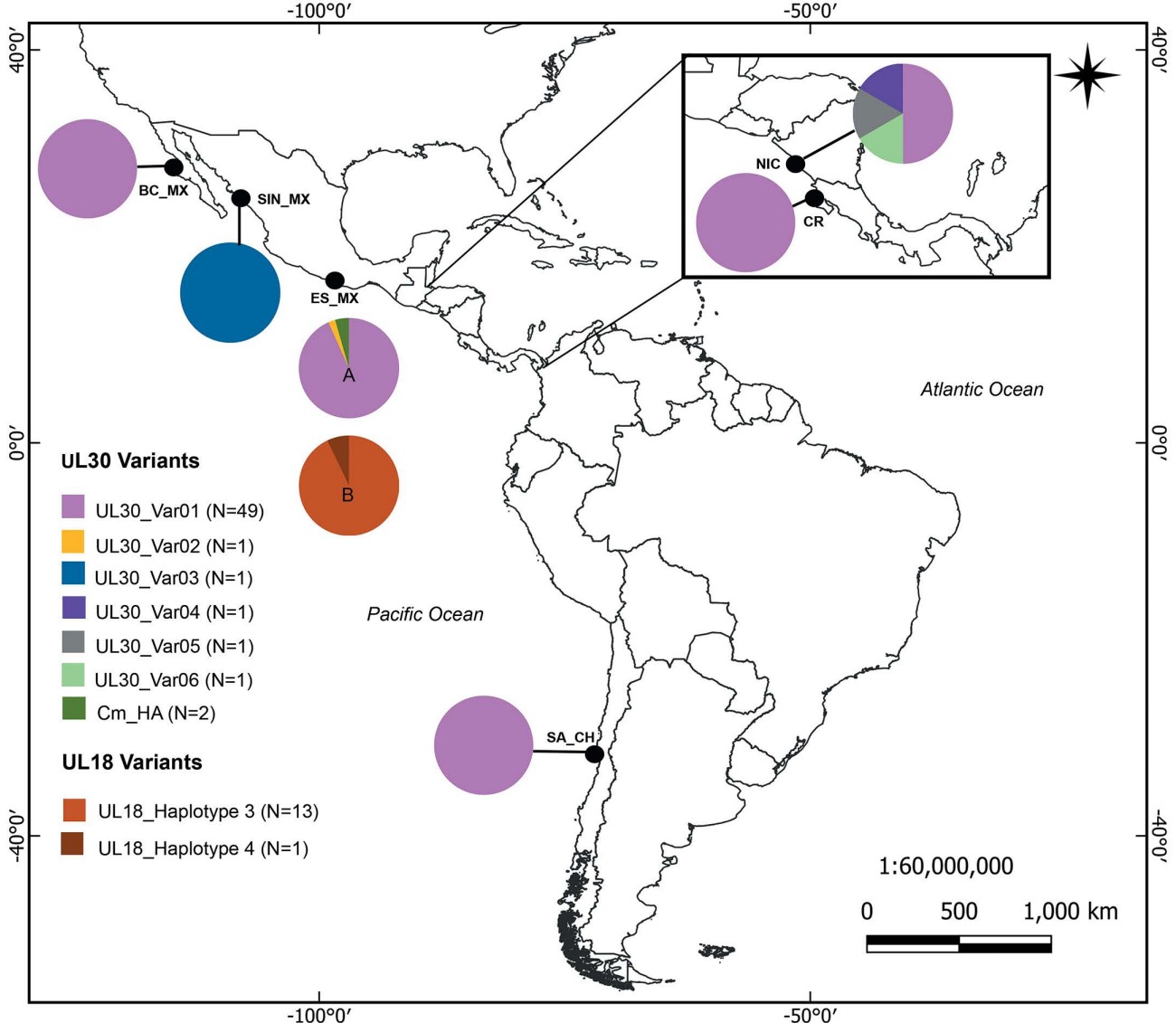

**Fig 4. Geographic distribution and frequency of ChAHV5 genetic variants based on the UL30 (A) and UL18 (B) genes in olive ridley turtles from the Pacific Ocean.** Pie charts represent the proportion of genetic variants at each locality. Abbreviations of localities: BC_MX, Baja California Sur, Mexico; SIN_MX, Sinaloa, Mexico; ES_MX, Playa Escobilla, Mexico; SA_CH, San Antonio, Chile; NIC, Nicaragua; CR, Costa Rica. The number in parentheses indicates the *N* sequences corresponding to each genetic variant. Base map from OpenStreetMap contributors (https://www.openstreet-map.org), licensed under the Open Database License (ODbL), via FAO Geospatial Platform (https://data.apps.fao.org/).

In contrast to the stable FP prevalence observed in nesting olive ridley females at Playa Escobilla, variable prevalence rates have been reported over time at Ostional, Costa Rica. In 1998, two surveys conducted in January (300 examined females) and February (125 examined females) reported FP prevalences of 4% and 10%, respectively [4]. In 2005, a lower prevalence of 1.2% was recorded (2,054 examined turtles), whereas a long-term study conducted between 2010 and 2013 showed an increase to 14% (74 examined turtles) [31,33]. Despite poor water quality, a high nitrogen footprint, and chemical and biological contamination [29], our findings indicate that FP values remain consistently low over time, at least along the Mexican Pacific coast.

The tumors on the body surface of females at Playa Escobilla were distributed in soft tissue, most of them in the forelimbs, shoulders, and neck. Similar results were previously reported in the same host and locality [32,40], suggesting an occurrence bias toward these body surfaces. In this study, we recorded only one case of an olive ridley female with a small tumor (length 0.9 cm) in the eye. Tumors on the eyelids and conjunctiva have been most frequently reported in juvenile green turtles [53].

The severity of FP cases in olive ridley females from Playa Escobilla was mostly mild, with 51 of 62 affected turtles within this category. In contrast, only eight and three individuals were categorized as moderate and severe cases, respectively. Using slightly different methodologies, most bases on tumor size, similar results were reported in the same host and locality in 2006 by Reséndiz et al. [40] and olive ridley females from a solitary nesting beach in Baja California, Mexico [34]. Vega-Hernández et al. [34] recorded eight FP-affected females at a solitary nesting beach in Baja California, Mexico of which six were classified as mild, one as moderate, and one as severe. Only one female presented a lesion larger than 10 cm, unlike the nine such cases found at Playa Escobilla. This difference could result from the high nesting female density at Playa Escobilla, which increases the probability of encountering FP-affected turtles with varying degrees of severity. Also, anthropogenic stressors, environmental pollutants, habitat quality, the immune response, and genetic predisposition are suggested as factors that could influence tumor growth and disease severity in sea turtle populations [29,54].

## Viral DNA detection in nesting olive ridley females

The detection of ChAHV5 by amplifying UL30 from directly biopsied tumor DNA reached 100% (18 of 18 samples), and the same percentage was observed when analyzing healthy tissues from individuals with FP in flipper biopsies (2 of 2). Furthermore, ChAHV5 detection in tissue from clinically healthy turtles reached 91% (20 of 22). In the case of amplification of UL18, lower detection rates were observed across all three types of samples: 67% for tumoral samples, 0% for healthy flipper tissues dissected from individuals with FP, and 9% for clinically healthy turtles. Clearly, the use of UL30 is the good standard for ChAHV5 detection, given its ability to amplify viral DNA even from clinically healthy specimens.

These results open the discussion about the underestimation of FP prevalence when considering only individuals with external clinical signs, rather than incorporating molecular techniques for ChAHV5 detection in asymptomatic individuals [23,38,55]. The detection of viral DNA in individuals without any external sign of the viral infection may reflect the presence of tumor in internal organs or alternatively, be related to the pathogenesis of ChAHV5 [1]. Herpesviruses, including alphaherpesviruses, exhibit a biphasic life cycle comprising an acute phase or primary infection, followed by a lifelong persistent infection in the host or latent phase [56]. During the latent phase, the viral genome persists in the host cell nucleus, limiting viral proteins expression to avoid detection by the immune system [57]. However, ChAHV5 can reactivate when the immune response is reduced due to environmental factors, stress, or poor health conditions. In this sense, comprehensive reviews on FP have emphasized its multifactorial nature, suggesting that it is better described as a syndrome rather than a single disease [29,58].

## Detection of ChAHV5 in marine leeches and their relationship with the transmission of FP

The role of *Ozobranchus* leeches (*O. margoi* and *O. branchiatus*) as mechanical vectors of the virus has been proposed before due to the high viral loads in these annelids compared to other epibionts of sea turtles [21,22]. In total, we screened 35 *O. branchiatus* specimens for ChAHV5 using the UL30 locus, and only two were positive (6%). Of all the leech samples collected, only one specimen was collected from a tumoral lesion, however, it tested negative for ChAHV5. A previous study in Baja California, Mexico, found viral DNA in a leech *O. branchiatus*, obtained from a ChAHV5-negative black turtle *Chelonia mydas agassizii*, Bocourt, 1868 [38].

The molecular detection rate of ChAHV5 in leeches varies widely among studies, ranging from as low as 6% (as reported in this study) to over 90% in leeches collected from juvenile green turtles in Florida [1]. In this sense, the

molecular detection of ChAHV5 alone cannot be conclusive evidence for a mechanical vector role for leeches. The low detection rate of ChAHV5 in leech samples using UL30 locus could be explained by the low viral DNA copy number in their tissue, making it difficult to detect using conventional PCR [15]. The metagenomic approach could provide more information about the virome of *Ozobranchus* leeches, as large-scale metagenomic studies of invertebrates have uncovered novel virus families and have even detected viral lineages previously thought to be restricted to vertebrates [59].

**Phylogeny and Identification of ChAHV5 genetic variants**

Both phylogenetic analyses, parsimony and ML, based on UL18 and UL30 resulted in poorly topologies with polytomies in most part of trees. These results were expected given the low variation in booth markers, with only 9 and 37 parsimony-informative characters out of 120 and 436, respectively. However, both phylogenetic trees resulting from the analyses of UL18 alone recovered samples from nesting olive ridley females in two parts of the tree, one of them include most of the samples (n = 13) and one sample with uncertain affinities with ChAHV5 of other sea turtles. Similar results were obtained from the UL30 gene analysis with one group of samples from olive ridley turtles and two samples (one from turtle and one from leech), more closely related to ChAHV5 isolated from green turtle.

We identified ChAHV5 variants by comparing our newly generated sequences with published sequences of the UL18 and UL30 genes. Based on the capsid protein gene fragment (UL18), a novel genetic variant, named UL18_haplotype 3, was identified in this study and occurs at high frequency exclusively in olive ridley turtles from Playa Escobilla. Alfaro-Núñez et al. [23] identified two genetic variants for this locus: haplotype 1 and haplotype 2. Haplotype 1 was identified at high frequency in samples of FP-affected turtles across multiple species (green, hawksbill, and leatherback) and was distributed in several regions (Hawaii, Island Principe, and Costa Rica) [23], while haplotype 2 was found in green turtles from the Atlantic, primarily in the Turks and Caicos Islands and along the northeastern coast of Brazil [23,55]. The remaining UL18 variants, haplotypes 4 (GenBank accession number OQ189658) and 5 (GenBank accession number KY933583), have been reported in specific geographical areas [25,60], this could indicate their restricted occurrence in the Pacific and Taiwan, respectively. Viral capsid formation is linked to the cleavage and packaging of the viral genome [61]. Consequently, alterations in the UL18 gene's protein products result in viral particles that replicate DNA but fail to cleave and package it, thereby restricting the production of viable virions. [61]. Also, it has been proposed that modifications to the viral capsid protein structure could result in differential virulence or increased resistance to the host immune system [23].

For UL30 gene, six genetic variants were identified in olive ridley turtles and *O. branchiatus* leeches. The most frequent variant was UL30_Var01 (GenBank accession number AF049904) [10], which was widely distributed from Baja California, Mexico, to the coast of San Antonio, Chile [33,35,38]. This variant was the most abundant in the olive ridley samples, as well as in an *Ozobranchus* leech collected in Playa Escobilla. In some alphaherpesviruses, UL30 gene exhibits high genetic variability, enabling the identification of novel variants, as seen in Herpes Simplex Virus 2 (HSV-2) [62]. Although DNA polymerase is a conserved gene essential for viral DNA replication and maintenance, it also contains hypervariable regions that serve as specific molecular signatures, facilitating the identification of viral variants [62].

Our study provides an updated assessment of fibropapillomatosis in the olive ridley nesting population from Playa Escobilla, one of the world's largest mass-nesting rookeries, revealing a persistently low prevalence of the disease over the past two decades. The high detection rate of ChAHV5 using molecular markers—even in clinically healthy females—suggests latent viral persistence within the population and underscores the importance of continuous monitoring. Although the presence of ChAHV5 in leeches was confirmed, low detection frequency does not support its role as viral vectors. Furthermore, our study identified a novel ChAHV5 variant exclusive to olive ridley turtles (UL18_haplotype 3), along with multiple UL30 variants broadening our understanding of the viral genetic diversity in the eastern Pacific. Overall, these findings highlight the need to integrate molecular surveillance with ecological monitoring at nesting sites to better elucidate the mechanisms underlying FP transmission and pathogenesis, while addressing key challenges for the conservation and management of threatened sea turtle populations.

## Supporting information

**S1 Table. Estimated number of nesting females during the past three nesting seasons at Playa Escobilla, Oaxaca, Mexico.** Number of females per nesting season was estimated by dividing the average number of nests by the clutch frequency (using a mean value of 2.5 nests per female per season, according to Abreu-Grobois and Plotkin, 2008).
(PDF)

**S2 Table. PCR Primers used for detection of chelonid alphaherpesvirus 5 (ChAHV5).**
(PDF)

**S3 Table. Sequences available in GenBank and Dryad for the partial UL18 gene (~110 bp) from different sea turtle species and localities.** Country name abbreviations are as follows: AF = Africa, BR = Brazil, CR = Costa Rica, DNK = Denmark, KWT = Kuwait, MY = Malaysia, MX = Mexico, U.S. = United States, TWN = Taiwan.
(XLSX)

**S4 Table. Sequences available in GenBank for the partial UL30 gene (~430 bp) from different sea turtle species and localities.** Country name abbreviations are as follows: BR = Brazil, CH = Chile, MX = Mexico, PR = Puerto Rico, U.S. = United States.
(XLSX)

**S5 Table. Summary of biological and clinical data from 62 olive ridley females affected by fibropapillomatosis at Playa Escobilla, Oaxaca, Mexico during nesting season 2022–2023.**
(XLSX)

**S6 Table. Variable sites and genetic variants for the partial UL18 gene (~110 bp) identified in ChAHV5 sequences isolated from several sea turtle species.**
(XLSX)

**S7 Table. Variable sites and genetic variants for the partial UL30 gene (~435 bp) identified in ChAHV5 sequences isolated from olive ridley turtles.** Country name abbreviations are as follows: U.S. = United States, MX = Mexico.
(XLSX)

**S1 Fig. Parsimony strict consensus tree based on the UL30 gene of ChAHV5.** Sequences isolated from olive ridley turtles at Playa Escobilla are indicated in green (sample group is indicated by A = FP-affected turtles, B = healthy tissue of FP turtles, and C = clinically healthy turtles) and from *Ozobranchus branchiatus* leeches in blue. Bootstrap values (≥50%) are indicated at the respective nodes. The tree is rooted to Testudine alphaherpesvirus 3 (GenBank accession number NC027916). For sequences retrieved from GenBank, the accession numbers are provided in parentheses. Localities' abbreviations are BR = Brazil, MX = Mexico, PR = Puerto Rico, U.S. = United States, WI = West Indies.
(PDF)

**S2 Fig. Parsimony strict consensus tree based on the UL18 gene of ChAHV5.** Sequences obtained from olive ridley turtles at Playa Escobilla are indicated in green (sample group is indicated by A = FP-affected turtles and C = clinically healthy turtles). Bootstrap values (≥50%) are indicated at the respective nodes. The tree is rooted with Testudine alphaherpesvirus 3 (GenBank accession number NC027916). For sequences retrieved from GenBank or the Dryad repository, the accession numbers or ID are provided in parentheses. Localities' abbreviations are AF = Africa, BR = Brazil, CR = Costa Rica, DNK = Denmark, KWT = Kuwait, MY = Malaysia, MX = Mexico, TWN = Taiwan, U.S. = United States.
(PDF)

## Acknowledgments

We want to thank the Comisión Nacional de Áreas Naturales Protegidas (CONANP) and the Centro Mexicano de la Tortuga (CMT) for the facilities provided for conducting this study. We are particularly grateful to M. Harfush Melendez, R. Villanueva Alcazar, and G. González Padilla. Additionally, we would like to thank the students and volunteers of Universidad del Mar for their support in the fieldwork and sampling, with special thanks to D. Vásquez Sibaja for the laboratory processing of leech samples. Special thanks to Willi Hennig Society for subsidizing the program TNT and making it free available. We are thankful to A. Jiménez, L. Márquez-Valdelamar, and N. M. López-Ortiz of the Laboratorio Nacional de la Biodiversidad, Instituto de Biología, Universidad Nacional Autónoma de México (LANABIO-IB, UNAM) for their assistance in laboratory and sequencing procedures. We thank the OpenStreetMap contributors for providing the map data used in Fig 4, available from https://www.openstreetmap.org under the Open Database License (ODbL; https://www.openstreet-map.org/copyright). We thank Dirección General de Vida Silvestre (DGVS) and to the Secretaría de Medio Ambiente y Recursos Naturales (SEMARNAT) for issuing the sample collection permit.

## Author contributions

**Conceptualization:** Elizabeth Labastida-Estrada, Karina Marisol Lugo-Trejo, Valentina Islas-Villanueva, F. Alberto Abreu-Grobois, Alejandro Oceguera-Figueroa.

**Data curation:** Elizabeth Labastida-Estrada, Karina Marisol Lugo-Trejo.

**Formal analysis:** Elizabeth Labastida-Estrada, Karina Marisol Lugo-Trejo, F. Alberto Abreu-Grobois, Alejandro Oceguera-Figueroa.

**Funding acquisition:** Valentina Islas-Villanueva, Alejandro Oceguera-Figueroa.

**Investigation:** Elizabeth Labastida-Estrada, Karina Marisol Lugo-Trejo.

**Methodology:** Elizabeth Labastida-Estrada, Karina Marisol Lugo-Trejo, Valentina Islas-Villanueva, Francisco Benítez-Villalobos, F. Alberto Abreu-Grobois.

**Project administration:** Valentina Islas-Villanueva, Alejandro Oceguera-Figueroa.

**Resources:** Valentina Islas-Villanueva, Francisco Benítez-Villalobos, Alejandro Oceguera-Figueroa.

**Software:** Elizabeth Labastida-Estrada.

**Supervision:** Valentina Islas-Villanueva, Francisco Benítez-Villalobos, F. Alberto Abreu-Grobois, Alejandro Oceguera-Figueroa.

**Validation:** Elizabeth Labastida-Estrada, F. Alberto Abreu-Grobois, Alejandro Oceguera-Figueroa.

**Visualization:** Elizabeth Labastida-Estrada.

**Writing – original draft:** Elizabeth Labastida-Estrada.

**Writing – review & editing:** Elizabeth Labastida-Estrada, Karina Marisol Lugo-Trejo, Valentina Islas-Villanueva, Francisco Benítez-Villalobos, F. Alberto Abreu-Grobois, Alejandro Oceguera-Figueroa.

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
