## [Decision Letter · Decision Letter 0]

1 Oct 2025

Dear Dr. Oceguera-Figueroa,

Thank you for submitting your manuscript to PLOS ONE. After careful consideration, we feel that it has merit but does not fully meet PLOS ONE’s publication criteria as it currently stands. Therefore, we invite you to submit a revised version of the manuscript that addresses the points raised during the review process.

We look forward to receiving your revised manuscript.

Kind regards,

Luisa Maria Diele-Viegas, Ph. D.

Academic Editor

PLOS ONE

Journal Requirements:

2. To comply with PLOS One submissions requirements, in your Methods section, please provide additional information regarding the experiments involving animals and ensure you have included details on (1) methods of sacrifice, (2) methods of anesthesia and/or analgesia, and (3) efforts to alleviate suffering.

3. Thank you for stating in your Funding Statement: [This project was partially funded by the Programa de Apoyo a Proyectos de Investigación e Innovación Tecnológica IN215722 of the Universidad Nacional Autónoma de México (PAPIIT-UNAM). The Dirección General de Asuntos del Personal Académico of the Universidad Nacional Autónoma de México (DGAPA-UNAM) provided financial support to E.L.E. through a postdoctoral fellowship. The Secretaría de Ciencias, Humanidades, Tecnología e Innovación (SECIHTI) supplied a grant scholarship (No. 812903) to K.M.L.T. (No. CVU: 1165342) and financial support through the Investigadoras e Investigadores por México program to V.I.V. (No. project 538).].

4. Thank you for stating the following financial disclosure: [This project was partially funded by the Programa de Apoyo a Proyectos de Investigación e Innovación Tecnológica IN215722 of the Universidad Nacional Autónoma de México (PAPIIT-UNAM). The Dirección General de Asuntos del Personal Académico of the Universidad Nacional Autónoma de México (DGAPA-UNAM) provided financial support to E.L.E. through a postdoctoral fellowship. The Secretaría de Ciencias, Humanidades, Tecnología e Innovación (SECIHTI) supplied a grant scholarship (No. 812903) to K.M.L.T. (No. CVU: 1165342) and financial support through the Investigadoras e Investigadores por México program to V.I.V. (No. project 538).]. 

7. We note that Figure 4 in your submission contain [map/satellite] images which may be copyrighted. All PLOS content is published under the Creative Commons Attribution License (CC BY 4.0), which means that the manuscript, images, and Supporting Information files will be freely available online, and any third party is permitted to access, download, copy, distribute, and use these materials in any way, even commercially, with proper attribution. For these reasons, we cannot publish previously copyrighted maps or satellite images created using proprietary data, such as Google software (Google Maps, Street View, and Earth). For more information, see our copyright guidelines: http://journals.plos.org/plosone/s/licenses-and-copyright.

1. You may seek permission from the original copyright holder of Figure 4 to publish the content specifically under the CC BY 4.0 license. 

Reviewers' comments:

Reviewer's Responses to Questions

**Comments to the Author**

1. Is the manuscript technically sound, and do the data support the conclusions?

Reviewer #1: Yes

Reviewer #2: Yes

2. Has the statistical analysis been performed appropriately and rigorously?

Reviewer #1: Yes

Reviewer #2: N/A

3. Have the authors made all data underlying the findings in their manuscript fully available?

Reviewer #1: Yes

Reviewer #2: Yes

4. Is the manuscript presented in an intelligible fashion and written in standard English?

Reviewer #1: Yes

Reviewer #2: Yes

Reviewer #1: The article entitled “Prevalence, etiology, and transmission of fibropapillomatosis in Olive Ridley turtles at a mass-nesting colony in the Mexican Pacific” is well-written and presents a sound methodology. The study successfully addressed all its objectives, which is essential for a high-quality scientific paper.

My main concern with the manuscript is the use of a pie chart, which could be replaced with a bar chart, for instance, to improve clarity. Additionally, the reference list includes a large number of sources published more than ten years ago and relatively few from the past five years. I recommend that the authors revise the bibliography and consider removing outdated references that are not strictly necessary.

Finally, I congratulate the authors on this valuable contribution to sea turtle conservation.

Reviewer #2: The manuscript is well-written, with an appropriate methodology and a solid discussion. However, some methodological points need to be detailed, and a few parts of the discussion could be improved.

ABSTRACT

The abstract is excellent; it just needs a final concluding sentence to summarize the main finding.

INTRODUCTION

The introduction is a bit long, but I would not change it because it contains a lot of interesting information that helps the reader understand the study's scope. In lines 148 to 153, it would be beneficial to add the prevalence of the disease in the mentioned locations, so the reader can understand whether the prevalence is high or low in this species.

MATERIAL AND METHODS

The Materials and Methods section is good but requires more detail in some parts.

1- In lines 189-192, please cite Table 1, which contains the estimated population size. I felt the absence of this information at this point.

2- Please explain in more detail how the beach was monitored during collections. Was it monitored every day? What times were the collections made? Was the patrol done on foot or with a vehicle (e.g., ATV)? How many people participated in the monitoring?

3- What specific non-toxic paint was used? Were the animals also tagged, or were they only identified with the paint?

4- From what part of the body was the tissue sample obtained (e.g., neck, flipper)?

5- How was the blood collected? Was it during or after nesting? Was an anticoagulant used? In what type of tube was the blood stored after collection?

6- How were the tumors collected? Was the entire tumor collected, or just a piece? Was there a preference for smaller tumors, which the literature suggests have a higher viral load?

7- What is the prevalence of leeches on the turtles? Can you estimate this, or at least indicate if they are commonly found on the turtles or if they are rare? This information would be interesting to the discussion (e.g., viral transmission).

8- Was only one leech collected from each animal, or were more than one collected?

RESULTS

Congratulations on the large sample size of turtles examined. The effort you made is commendable, even with a large population in the area.

1- In lines 320-321, it says 22 tissue samples, with 9 being blood and 14 being tissue (9 + 14 = 23). Please double-check this information.

DISCUSSION

The discussion needs some adjustments.

1- Line 451: What was the sample size of the 1988 study? Including this information is important to understand how many animals were sampled and to be able to confidently state that the prevalence has decreased.

2- Lines 453-455: Was the site monitored before this? It would be good to have information on how long this population has been monitored to be able to state that there were no cases before then.

3- Lines 464-468: The comparison made here is unfair, as it is known that green sea turtles are the most affected species and therefore have the highest prevalence. Additionally, juveniles also have a much higher prevalence than adults. Either remove this part of the discussion or compare it with other nesting areas of the olive ridley turtle. If you don't find many studies on olive ridleys, try comparing them with other species, but from nesting areas, not foraging areas. Only then will you be able to support the information in lines 470-472.

4- Lines 479-480: Are the green sea turtles mentioned here juveniles? If so, it is more relevant to compare them with adults of the same species or, if there are no articles available, with adults of other species.

5- Lines 487-490: Again, are the green sea turtles juveniles? It's always good to make this information clear because it makes a big difference. The discussion would be much better and interesting if it compared between species in nesting areas.

6- Lines 501-502: What are the "other tissues" (2 of 2)?

7- Line 507: There is a typo: "god" standard should be "good" standard.

8- Topic "viral DNA detection in nesting olive ridley females": What do these findings reveal, considering the prevalence of fibropapillomatosis found? Even with a low sample size of tissue from healthy turtles, the prevalence of the virus was high. Could this be related to the environment they live in or the animals' health?

9- Line 536: With which turtle species and in which location was the study conducted that found 90% viral detection in leeches?

10- Could the low viral detection also be related to the small number of leeches sampled?

11- I missed a final concluding paragraph summarizing the main findings.

**Do you want your identity to be public for this peer review?** For information about this choice, including consent withdrawal, please see our Privacy Policy

Reviewer #1: No

Reviewer #2: **Yes: ** Camila Miguel

---

## [Author Response · Author response to Decision Letter 1]

4 Nov 2025

RESPONSES TO REVIEWERS

Manuscript reference: PONE-D-25-37483

Prevalence, etiology, and transmission of fibropapillomatosis in Olive Ridley turtles at a mass-nesting colony in the Mexican Pacific

Luisa Maria Diele-Viegas, Ph. D.

Academic Editor

PLOS ONE

Please find in this letter our responses to the reviewers’ comments. All changes in the manuscript that correspond to responses to Reviewer # 1 are in green type, those to Reviewer #2 are in red type. Additionally, responses to the Journal requirements are highlighted in purple. If you have any questions, please feel free to contact us for additional explanation.

Journal Requirements:

Yes, we verify that our manuscript meets all style and file naming requirements.

2. To comply with PLOS One submissions requirements, in your Methods section, please provide additional information regarding the experiments involving animals and ensure you have included details on (1) methods of sacrifice, (2) methods of anesthesia and/or analgesia, and (3) efforts to alleviate suffering.

Yes, we confirm that our manuscript meets these requirements. In addition to the statements regarding legal permits for sampling activities (Lines 181-185 in the revised version of the manuscript), we included information on the efforts made to minimize discomfort and stress to the individuals, as well the sanitization measures applied to reduce the risk of infection from the tissue sampling (Lines 232-235 in the revised version of the manuscript). In our study, no animals were sacrificed or injured to the degree that required anesthesia or analgesia.

3. Thank you for stating in your Funding Statement: [This project was partially funded by the Programa de Apoyo a Proyectos de Investigación e Innovación Tecnológica IN215722 of the Universidad Nacional Autónoma de México (PAPIIT-UNAM). The Dirección General de Asuntos del Personal Académico of the Universidad Nacional Autónoma de México (DGAPA-UNAM) provided financial support to E.L.E. through a postdoctoral fellowship. The Secretaría de Ciencias, Humanidades, Tecnología e Innovación (SECIHTI) supplied a grant scholarship (No. 812903) to K.M.L.T. (No. CVU: 1165342) and financial support through the Investigadoras e Investigadores por México program to V.I.V. (No. project 538).].

We made changes to the Funding Statement and provided an amended version with the requested details.

4. Thank you for stating the following financial disclosure: [This project was partially funded by the Programa de Apoyo a Proyectos de Investigación e Innovación Tecnológica IN215722 of the Universidad Nacional Autónoma de México (PAPIIT-UNAM). The Dirección General de Asuntos del Personal Académico of the Universidad Nacional Autónoma de México (DGAPA-UNAM) provided financial support to E.L.E. through a postdoctoral fellowship. The Secretaría de Ciencias, Humanidades, Tecnología e Innovación (SECIHTI) supplied a grant scholarship (No. 812903) to K.M.L.T. (No. CVU: 1165342) and financial support through the Investigadoras e Investigadores por México program to V.I.V. (No. project 538).].

We made changes to the Funding Statement and provided an amended version with the requested details.

This project was partially funded by the Programa de Apoyo a Proyectos de Investigación e Innovación Tecnológica IN215722 of the Universidad Nacional Autónoma de México (PAPIIT-UNAM). The Dirección General de Asuntos del Personal Académico of the Universidad Nacional Autónoma de México (DGAPA-UNAM) provided financial support to E.L.E. through a postdoctoral fellowship. The Secretaría de Ciencias, Humanidades, Tecnología e Innovación (SECIHTI) supplied a grant scholarship (No. 812903) to K.M.L.T. (No. CVU: 1165342) and financial support through the Investigadoras e Investigadores por México program to V.I.V. (No. project 538).

We have made the requested changes

All genetic sequences generated in this study have been deposited in the DNA Data Bank of Japan (DDBJ) under accession numbers LC899718–LC899775 and LC899376–LC899383. These sequences will be publicly available through the DDBJ website within 2 to 7 days from November 4, 2025, once released by DDBJ. The Data Availability Statement in the manuscript has been updated accordingly to reflect this information.

7. We note that Figure 4 in your submission contain [map/satellite] images which may be copyrighted. All PLOS content is published under the Creative Commons Attribution License (CC BY 4.0), which means that the manuscript, images, and Supporting Information files will be freely available online, and any third party is permitted to access, download, copy, distribute, and use these materials in any way, even commercially, with proper attribution. For these reasons, we cannot publish previously copyrighted maps or satellite images created using proprietary data, such as Google software (Google Maps, Street View, and Earth). For more information, see our copyright guidelines: http://journals.plos.org/plosone/s/licenses-and-copyright.

1. You may seek permission from the original copyright holder of Figure 4 to publish the content specifically under the CC BY 4.0 license.

Figure 4 was created specifically for this study and is therefore not subject to copyright restrictions. The map was produced using QGIS v.3.16 (free and open-source software). The base layer was obtained from OpenStreetMap contributors (available from https://www.openstreetmap.org), licensed under the Open Database License (ODbL; https://www.openstreetmap.org/copyright), and accessed through the FAO Geospatial Platform (https://data.apps.fao.org/). The figure caption now includes proper attribution to OpenStreetMap and clarifies that the data are available under the ODbL license, as well as in Acknowledgments section (Lines 469-471 and 649-6532 in the revised version of the manuscript).

This comment does not apply to our case

We have thoroughly reviewed the reference list and would like to note the following:

We did not find any retracted articles in our reference list.

We removed outdated references and those not strictly necessary, as suggested by Reviewer #1. As a result, the reference numbers have changed in this revised version of the manuscript.

Review Comments to the Author

Reviewer #1

The article entitled “Prevalence, etiology, and transmission of fibropapillomatosis in Olive Ridley turtles at a mass-nesting colony in the Mexican Pacific” is well-written and presents a sound methodology. The study successfully addressed all its objectives, which is essential for a high-quality scientific paper.

1. (A) My main concern with the manuscript is the use of a pie chart, which could be replaced with a bar chart, for instance, to improve clarity. (B) Additionally, the reference list includes a large number of sources published more than ten years ago and relatively few from the past five years. I recommend that the authors revise the bibliography and consider removing outdated references that are not strictly necessary.

Comment 1A. We appreciate Reviewer #1 suggestion to replace the pie chart in Figure 4 with a bar chart. Nonetheless, pie charts are widely used to illustrate haplotype composition and genetic variant proportions in population studies. We explored the use of a bar chart for Figure 4, but it was less effective in conveying the data and did not improve interpretability; therefore, we opted to retain the pie chart format.

Comment 1B. We removed outdated references and those not strictly necessary, as suggested. As a result, the reference numbers have changed in this revised version of the manuscript.

Finally, I congratulate the authors on this valuable contribution to sea turtle conservation.

Reviewer #2

The manuscript is well-written, with an appropriate methodology and a solid discussion. However, some methodological points need to be detailed, and a few parts of the discussion could be improved.

ABSTRACT

1. The abstract is excellent; it just needs a final concluding sentence to summarize the main finding.

We have included a final sentence summarizing our key findings (Lines 43-46 in the revised version of the manuscript).

INTRODUCTION

2. The introduction is a bit long, but I would not change it because it contains a lot of interesting information that helps the reader understand the study's scope. In lines 148 to 153, it would be beneficial to add the prevalence of the disease in the mentioned locations, so the reader can understand whether the prevalence is high or low in this species.

We have now included the FP prevalence data corresponding to the nesting beaches mentioned in lines 148–153 original version of MS (Lines 151-156 in the revised version of the manuscript).

MATERIAL AND METHODS

The Materials and Methods section is good but requires more detail in some parts.

3. In lines 189-192, please cite Table 1, which contains the estimated population size. I felt the absence of this information at this point.

In lines 189–192, we clarified that the sampling effort was estimated based on the population size from the three previous nesting seasons (2019–2020, 2020-2021, and 2021-2022), whereas Table 1 presents the estimated female population size per arribada for the nesting season during this study (2022–2023). To fully address the reviewer’s request, we have now included this data in a supplementary table (Table S1, Line 197 in the revised version of the manuscript).

4. Please explain in more detail how the beach was monitored during collections. Was it monitored every day? What times were the collections made? Was the patrol done on foot or with a vehicle (e.g., ATV)? How many people participated in the monitoring?

Night surveys were conducted every day along the nesting beach to detect FP cases in nesting females. Surveys were carried out on foot along the nesting area during the first five nights of each arribada, between 21:00 and 04:00 h. During these nights, we walked approximately 7 km, covering the main nesting zone. However, when nesting activity was concentrated in high-density areas, efforts were focused on those zones. Each survey was conducted by a team of four to five people, most of whom were volunteers. On some nights when volunteers were unavailable, only two or three people partici

---

## [Decision Letter · Decision Letter 1]

3 Dec 2025

Prevalence, etiology, and transmission of fibropapillomatosis in Olive Ridley turtles at a mass-nesting colony in the Mexican Pacific

PONE-D-25-37483R1

Dear Dr. Oceguera-Figueroa,

We’re pleased to inform you that your manuscript has been judged scientifically suitable for publication and will be formally accepted for publication once it meets all outstanding technical requirements.

Kind regards,

Luisa Maria Diele-Viegas, Ph. D.

Academic Editor

PLOS ONE

Additional Editor Comments (optional):

Reviewers' comments:

Reviewer's Responses to Questions

**Comments to the Author**

Reviewer #1: All comments have been addressed

Reviewer #2: All comments have been addressed

2. Is the manuscript technically sound, and do the data support the conclusions?

Reviewer #1: Yes

Reviewer #2: Yes

3. Has the statistical analysis been performed appropriately and rigorously?

Reviewer #1: Yes

Reviewer #2: N/A

4. Have the authors made all data underlying the findings in their manuscript fully available?

Reviewer #1: Yes

Reviewer #2: Yes

5. Is the manuscript presented in an intelligible fashion and written in standard English?

Reviewer #1: Yes

Reviewer #2: Yes

Reviewer #1: (No Response)

Reviewer #2: The manuscript was significantly improved after the review round. The authors provided detailed and satisfactory responses, and the modifications implemented completely addressed all concerns from the first assessment. I recommend Acceptance for Publication as it stands.

**Do you want your identity to be public for this peer review?** For information about this choice, including consent withdrawal, please see our Privacy Policy

Reviewer #1: No

Reviewer #2: **Yes: ** Camila Miguel

---

## [Editor Report · Acceptance letter]

PONE-D-25-37483R1

PLOS One

Dear Dr. Oceguera-Figueroa,

I'm pleased to inform you that your manuscript has been deemed suitable for publication in PLOS One. Congratulations! Your manuscript is now being handed over to our production team.

Kind regards,

on behalf of

Dr. Luisa Maria Diele-Viegas

Academic Editor

PLOS One